# Entanglement entropy with Lifshitz fermions

Dion Hartmann[1*], Kevin Kavanagh[2,3] and Stefan Vandoren[1]

**1** Institute for Theoretical Physics, Utrecht University,
Leuvenlaan 4, NL-3584 CE Utrecht, The Netherlands
**2** Dublin Institute for Advanced Studies, School of Theoretical Physics,
10 Burlington Road, Dublin 4, Ireland.
**3** Department of Theoretical Physics, Maynooth University,
Maynooth, Co. Kildare, Ireland.

⋆ d.m.f.hartmann@uu.nl

## Abstract

We investigate fermions with Lifshitz scaling symmetry and study their entanglement entropy in 1+1 dimensions as a function of the scaling exponent $z$. Remarkably, in the ground state the entanglement entropy vanishes for even values of $z$, whereas for odd values it is independent of $z$ and equal to the relativistic case with $z = 1$. We show this using the correlation method on the lattice, and also using a holographic cMERA approach. The entanglement entropy in a thermal state is a more detailed function of $z$ and $T$ which we plot using the lattice correlation method. The dependence on the even- or oddness of $z$ still shows for small temperatures, but is washed out for large temperatures or large values of $z$.

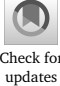
# 1 Introduction

In this paper, we study entanglement properties of Dirac-Lifshitz fermions, with dispersion relations of the form:

$$\omega_k^2 = \alpha^2 k^{2z} + m^2 \,, \tag{1}$$

with $\omega, k$ and $m$ related to frequency, momentum and mass, with units specified in the next section. Furthermore, $\alpha$ is a dimensionful constant and $z$ is a parameter, and we mostly consider cases where $z$ is an integer in order to avoid issues with branch cuts (e.g. when $z = 1/4$, negative $k$ would yield two branches). For $z = 1$, equation (1) yields the standard dispersion relation for a Dirac fermion with $\alpha = c$, the speed of light. We call $z$ the Lifshitz exponent, and for $m = 0$, the theory has Lifshitz scaling symmetry acting as:

$$\omega \to \Lambda^z \omega \,, \qquad \vec{k} \to \Lambda \vec{k} \,. \tag{2}$$

For this reason, they are called Lifshitz fermions. Besides the scale symmetry, there is rotation and translation symmetry and together with the scale symmetry they form the Lifshitz symmetry algebra. There are however no boost symmetries for $z \neq 1$, further discussions on symmetries can be found in e.g. [1–3]. Some earlier papers considered Lifshitz fermions with $z = 2$ and $z = 3$, see e.g. [4–6] in the context of the chiral anomaly, and [7, 8] where theories with four-fermi interactions are included. Experimentally, larger than expected dynamical exponents can be seen in heavy fermion systems [9, 10].

It is interesting to study properties of Lifshitz fermions as a function of the dynamical exponent $z$, and in this paper we will focus on correlation functions and entanglement entropy (EE), and in particular at the EE at the scale invariant point where $m = 0$. There is extensive literature on EE for free quantum field theories and lattice models with fermions. Various methods can be used, such as the correlation method in real space, the replica method, and Multi-scale Entanglement Renormalisation Ansatz (MERA). For a review see e.g. [11]. The strongest results exist for two-dimensional (1+1) relativistic conformal field theories, starting with the celebrated works of [12, 13]. For this reason, we focus on two dimensions in this paper as well, to see how the known results from relativistic CFTs change when changing the value of $z$ away from one. The holomorphic properties of relativistic CFTs do not, however, apply for $z \neq 1$, and the techniques therefore have to be adapted. We will use two techniques: the correlation method on the lattice [14–16], and the holographic cMERA approach [17].

On the lattice $z$ denotes the range of the interactions: $z = 1$ is nearest neighbor, $z = 2$ next-to-nearest and large values of $z$ imply longe range interactions as illustrated in figure 1. The lattice spacing breaks conformal invariance, but our numerics are accurate enough to be close to the continuum limit. Furthermore, on the lattice, one can study how the EE changes in the presence of long-range interactions.

Entanglement entropy for Lifshitz bosons also have been studied, such as in the quantum Lifshitz model with $z = 2$ in 2+1 dimensions (see e.g. [18–25] for a partial list of references), and more generally for $z = d + 1$ in [26–28]. More recently studies for generic $z$ were carried out in in e.g. [29–31], see also [32–34] for further references on related topics. The results for bosons compared to fermions differ quite a lot. For even values of $z$, the EE for massless fermions turns out to vanish in the ground state, whereas for bosons, it is nonzero. For odd values, the EE is independent of $z$, i.e. all odd values for $z$ give the same result as for $z = 1$. Again, this is very different from Lifshitz bosons, where the EE grows with $z$ as expected from the lattice approach, since higher values of $z$ indicate longer range correlators across the entanglement regions. For fermions, however, these correlations seem to cancel out in the EE. The distinction between even and odd values of $z$ is quite striking for fermions, and indicate that one cannot simply extrapolate to continuous values of $z$, at least not in an obvious way.

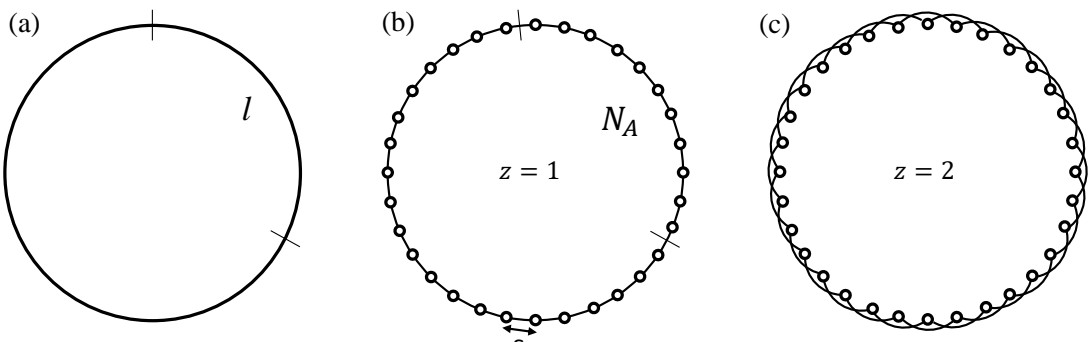

Figure 1: The continuum system (a) of length $L$ is partitioned in a segment of length $l$ and its complement. The lattice system (b and c) has $N$ sites and a lattice spacing $\varepsilon$. The interactions are depicted for $z = 1$ (b) and $z = 2$ (c).

This picture is also confirmed by the holographic cMERA approach [17], which nicely reproduces our results obtained from the lattice correlation method. The use of the holographic cMERA approach is therefore of independent interest, as was illustrated for Lifshitz scalar fields in [31].

At finite temperature, we generate EE also for even values of $z$. We study both the small and large temperature regimes on the lattice, and we show that the parity of $z$ (even or odd) does not play an important role anymore at high temperature.

This paper is organized as follows. In section 2 we introduce the basics, present the Lagrangian for free Lifshitz fermions and we determine the two-point correlator. We also review the exact results known for $z = 1$, and we make an ansatz for the EE for $z > 1$ using Lifshitz scale invariance at $m = 0$. In section 3, we discretize the model and compute the correlators on the lattice. We use the correlator method to compute the EE on the lattice and present various cases. In section 4, we rederive the zero temperature results using the cMERA approach for fermions. We end with some conclusions.

## 2  Lifshitz fermions in 1+1 dimensions

The Lagrangian for a two-component Lifshitz free fermion in two spacetime dimensions with coordinates $\{x^0, x^1\} = \{t, x\}$, is given by

$$\mathcal{L} = \bar{\psi}(\hbar\gamma^0 i\partial_0 + \hbar\alpha\gamma^1(i\partial_1)^z - \mu\alpha^2)\psi\,, \tag{3}$$

with $\bar{\psi} \equiv \psi^\dagger\gamma^0$, and Dirac matrices satisfying the Clifford algebra $\{\gamma^\mu, \gamma^\nu\} = 2\eta^{\mu\nu}\mathbb{I}_{2\times2}$. The path integral is then weighted with the standard factor $\exp(iS/\hbar)$ with $S = \int dt dx \,\mathcal{L}$. Here, $\alpha$ has SI-units $m^z/s$ and is the speed of light for $z = 1$, and $\mu$ has units $kg/m^{2z-2}$ and is the mass for $z = 1$. The units of $\psi$ are $m^{-1/2}$ and for $z = 1$ we recover the relativistic Dirac Lagrangian. The Lifshitz scale transformations reads

$$t \rightarrow \lambda^z t\,, \qquad x \rightarrow \lambda x\,, \qquad \psi \rightarrow \lambda^{-1/2}\psi \tag{4}$$

and is only a symmetry of the Lagrangian for $\mu = 0$. We will mostly consider the massless case in this paper. Notice that the scaling weight for a fermion is independent of $z$ (in any number of dimensions!), in contrast with a free boson, whose scaling weight is $(z-1)/2$. This fact has consequences for the EE which we discuss extensively in this paper.

Space time translation symmetry, together with the Lifshitz scale symmetry generate the Lifshitz algebra in 1+1 spacetime dimension. There is no boost symmetry for generic $z \neq 1$,

but there is a U(1) symmetry acting as an overall phase on $\psi$. In the massless case, there is also chiral symmetry.

Our conventions are as follows. With the $(1+1)$-dimensional metric $\eta = diag(+1,-1)$ we choose our Clifford algebra to be[1]

$$\gamma^0 = \begin{pmatrix} 0 & 1 \\ 1 & 0 \end{pmatrix}, \qquad \gamma^1 = \begin{pmatrix} 0 & -1 \\ 1 & 0 \end{pmatrix}. \tag{5}$$

Furthermore, we can define chiral components using

$$P_\pm = \frac{1 \pm \gamma^5}{2}, \qquad \gamma^5 \equiv \gamma^0 \gamma^1 = \begin{pmatrix} 1 & 0 \\ 0 & -1 \end{pmatrix}, \qquad \psi = \begin{pmatrix} \psi_+ \\ \psi_- \end{pmatrix}. \tag{6}$$

The Lagrangian then becomes

$$\mathcal{L} = \hbar \psi_+^\dagger \left(i\partial_0 + \alpha(i\partial_1)^z\right)\psi_+ + \hbar \psi_-^\dagger \left(i\partial_0 - \alpha(i\partial_1)^z\right)\psi_- - \mu\alpha^2 \left(\psi_+^\dagger \psi_- + \psi_-^\dagger \psi_+\right), \tag{7}$$

and only has chiral symmetry in the massless case, where $\psi_+$ and $\psi_-$ transform with opposite phases. One can easily check that the action is real upon partial integration. The equation of motion is

$$(i\slashed{D}_z - m)\psi \equiv \left(\gamma^0 i\partial_0 + \alpha\gamma^1(i\partial_1)^z - m\right)\psi = 0, \tag{8}$$

with $m \equiv \frac{\mu\alpha^2}{\hbar}$. Contrary to the $z = 1$ case, for $z \neq 1$, these chiralities do not correspond to left or right movers which is why all holomorphic CFT techniques no longer apply. With the plane wave ansatz

$$\psi(t,x) = \int d\omega \, dk \, \psi(\omega,k) e^{-i\omega t - ikx}, \tag{9}$$

one derives the Lifshitz dispersion relation (1). After Fourier transformation, the action becomes

$$S = (2\pi)^2 \int d\omega \, dk \, \psi^\dagger(\omega,k)\left(\hbar\omega + \hbar\alpha\gamma^5 k^z - \mu\alpha^2\gamma^0\right)\psi(\omega,k), \tag{10}$$

and the two-point correlator, for $t \equiv t_1 - t_2$ and $x \equiv x_1 - x_2$, is

$$\begin{aligned}
G_F(t,x) \equiv \langle\psi^{\dagger\alpha}(t_1,x_1)\psi_\beta(t_2,x_2)\rangle &= i\int_{-\infty}^{+\infty} \frac{d\omega}{2\pi}\frac{dk}{2\pi} \frac{(\omega - \alpha\gamma^5 k^z + m\gamma^0)^\alpha{}_\beta}{\omega^2 - \alpha^2 k^{2z} - m^2 + i\epsilon} e^{-i\omega t - ikx}, \\
&= \gamma^0\left((i\slashed{D}_z)^\dagger + m\right)G_B(t,x). \tag{11}
\end{aligned}$$

Here, the Lifshitz-scalar Green's function is given by

$$G_B(t,x) = i\int_{-\infty}^{+\infty} \frac{d\omega}{2\pi}\frac{dk}{2\pi} \frac{e^{-i\omega t - ikx}}{\omega^2 - \alpha^2 k^{2z} - m^2 + i\epsilon} = \int_{-\infty}^{+\infty} \frac{dk}{2\pi}\frac{e^{-ikx}}{2\omega_k}\left(e^{i\omega_k t}\theta(-t) + e^{-i\omega_k t}\theta(t)\right), \tag{12}$$

with $\omega_k = \sqrt{\alpha^2 k^{2z} + m^2}$ the positive root. Notice the usual relation with the propagator of a scalar field, this time a scalar field with a Lifshitz dispersion relation (1). The propagator $G_F(t,x)$ satisfies the Lifshitz-Dirac equation with a delta function source because of the identity

$$\left(i\slashed{D}_z - m\right)\gamma^0\left(i\slashed{D}_z^\dagger + m\right) = -\gamma^0\left(\partial_0^2 + \alpha^2(i\partial_1)^{2z} + m^2\right), \tag{13}$$

---

[1]In our basis, the charge conjugation matrix is chosen $\mathcal{C} = i\gamma^1$ satisfying $\mathcal{C}^\dagger = \mathcal{C}$, $\mathcal{C}^\dagger\mathcal{C} = 1$, and $\mathcal{C}\gamma^\mu\mathcal{C}^{-1} = -(\gamma^\mu)^T$. If we would impose the Majorana condition $\psi^\dagger\gamma^0 = \psi^T\mathcal{C}$, then it implies for the spinor components, $\psi_\pm^* = \mp i\psi_\pm$. The chiral Majorana components are not real, but this is because we are not in a basis with purely imaginary gamma matrices. The reality condition does respect the chiralities however, so $\psi_\pm$ each are Majorana-Weyl spinors.

and because the scalar field propagator satisfies the Lifshitz-Klein-Gordon equation for the Green's function. The $\gamma^0$ appears because we are considering the propagator $\langle \psi^\dagger \psi \rangle$ instead of $\langle \bar{\psi} \psi \rangle$.

It is interesting to look at the case of a free massless scalar field with Lifshitz scaling. For the equal time correlator, we get

$$\langle \phi(x_1)\phi(x_2) \rangle = \int\limits_{-\infty}^{+\infty} \frac{\mathrm{d}k}{2\pi} \frac{e^{-ikx}}{2\omega_k} = \frac{1}{2\pi\alpha} 2^{-z} \sqrt{\pi} \frac{\Gamma\left(\frac{1-z}{2}\right)}{\Gamma(\frac{z}{2})} |x|^{z-1}. \tag{14}$$

Notice that this is consistent with the scaling weight $(z-1)/2$ for a scalar field in 1+1 dimensions. The result for this Fourier transform is formally valid for all values of $z$ by analytic continuation of the Gamma function. If we restrict to integer values, we notice a difference between even and odd values of $z$, since $\Gamma\left(\frac{1-z}{2}\right)$ for even $z = 2n$ produces a factor $\Gamma(\frac{1}{2}-n) = \frac{(-4)^n n!}{(2n)!}\sqrt{\pi}$, whereas for odd $z = 2n+1$, we get $\Gamma(-n)$ which diverges as $\Gamma(z)$ has a simple pole at $z = -n$. In higher dimensions, a similar phenomena happens, as the higher dimensional Fourier transform produces factors of $\Gamma\left(\frac{d-z}{2}\right)$. This divergence needs to be regularized but we will not go further into this since it does not occur for fermions as we see now.

Similarly to the bosons, the fermionic two-point correlator is,

$$G_F(t,x) = \int\limits_{-\infty}^{+\infty} \frac{\mathrm{d}k}{2\pi} \frac{e^{-ikx}}{2\omega_k} \left[ e^{-i\omega_k t}\left(\omega_k - \alpha k^z \gamma^5 + m\gamma^0\right)\theta(t) - e^{i\omega_k t}\left(\omega_k + \alpha k^z \gamma^5 - m\gamma^0\right)\theta(-t) \right]. \tag{15}$$

We now focus on the massless case with $\omega_k = \alpha|k|^z$ where the chiral components decouple, and take the equal time correlator obtained from the limit $t \to 0^+$, to get

$$\langle \psi_\pm^\dagger(x_1)\psi_\pm(x_2) \rangle = \frac{1}{2} \int\limits_{-\infty}^{+\infty} \frac{\mathrm{d}k}{2\pi} e^{-ikx}\left(1 \mp \mathrm{sgn}(k)^z\right). \tag{16}$$

The result for the integral depends again on the even- or oddness of $z$:

$$z \text{ even}: \quad \langle \psi_+^\dagger(x_1)\psi_+(x_2) \rangle = 0, \quad \langle \psi_-^\dagger(x_1)\psi_-(x_2) \rangle = \delta(x_1 - x_2). \tag{17}$$

For odd values of $z$, we have $\mathrm{sgn}(k)^z = \mathrm{sgn}(k)$ and find

$$z \text{ odd}: \quad \langle \psi_\pm^\dagger(x_1)\psi_\pm(x_2) \rangle = \frac{1}{2}\left[ \delta(x_1 - x_2) \pm \frac{i}{\pi}\frac{1}{(x_1 - x_2)} \right], \tag{18}$$

independent of $z$. This independence of $z$ is consistent with the fact that the Lifshitz scaling weight for a fermion is independent of $z$ and equal to $-1/2$ in 1+1 dimensions. The expressions for the correlators are the two possibilities consistent with the Lifshitz symmetries with the correct scaling weight, as $\delta(\lambda x) = |\lambda|^{-1}\delta(x)$.

## 2.1 Entanglement entropy and relation to known results

What we learn from the analysis above in the continuum, is that at zero temperature and zero mass, the two-point function differs for even and odd values of $z$. In both classes, the correlator does not depend on $z$. So for odd $z$, the EE is the same as for the relativistic case with $z = 1$.

In that case, the result for the vacuum EE in a subinterval of length $l$ on the real infinite line is well known from conformal field theory, namely [12, 13, 35]

$$S = \frac{c}{3} \log\left(\frac{l}{\varepsilon}\right), \tag{19}$$

with $c = 1/2$ for a Weyl fermion and $\varepsilon$ the UV cutoff which is the lattice spacing in the next section. For even values of $z$, the spatial correlators produce zero or delta functions, and this will not produce any entanglement. We show this explicitly using the lattice model and the cMERA approach in subsequent sections.

We can consider finite size effects, and for a relativistic CFT on a line of total length $L$ and with periodic boundary conditions (see figure 1), we have [36]

$$S = \frac{c}{3} \log\left(\frac{L}{\pi \varepsilon} \sin\left(\frac{\pi l}{L}\right)\right), \tag{20}$$

up to some non-universal additive constant. This expression still obeys Lifshitz scaling properties, so it is a possible candidate for the Lifshitz EE for general values of $z$, but again only odd values. We show on the lattice that for odd values of $z$, the finite size effects do not depend on $z$, so we use the known results for $z = 1$. On a lattice with $N$ sites and a subsystem of $N_A$ sites, (20) becomes

$$S = \frac{c}{3} \log\left(\frac{N}{\pi} \sin\left(\frac{\pi N_A}{N}\right)\right). \tag{21}$$

For even values of $z$, finite size effects won't affect the spatial correlators as we show in the next section, so the EE still vanishes. Notice also the symmetry $N_A \to N - N_A$ which reflects one of the properties of EE in a pure state.

We now add temperature, still keeping $m = 0$ and $L \to \infty$. The result for the EE should still obey Lifshitz scale invariance, provided we scale the temperature appropriately, $T \to \lambda^{-z} T$. The only scale invariant and dimensionless quantities are

$$\frac{l}{\varepsilon}, \quad \frac{\varepsilon^z k_B T}{\alpha \hbar} \equiv \frac{\varepsilon^z}{\beta}, \tag{22}$$

and combinations thereof such as the cutoff independent quantity $l\beta^{-1/z}$. For $z = 1$ the result for the EE is known [36] and is given by

$$S = \frac{c}{3} \log\left(\frac{\beta}{\pi \varepsilon} \sinh\left(\frac{\pi l}{\beta}\right)\right). \tag{23}$$

This result holds when the system is infinitely long and in a thermal state.

At low temperatures, we obtain from (23),

$$S = \frac{c}{3}\left[ \log\left(\frac{l}{\varepsilon}\right) + \frac{\pi^2 l^2}{6\beta^2} + \mathcal{O}(l^4/\beta^4) \right], \qquad l \ll \beta, \tag{24}$$

consistent with the scaling properties for $z = 1$, for which $l/\beta$ is scale invariant. Notice that a linear term proportional to $l/\beta$ is absent in this Taylor expansion. Such a term would produce a volume law, which is what we expect at high temperatures. Indeed, the high temperature regime computed from (23) yields

$$S = \frac{c}{3}\left( \pi \frac{l}{\beta} - \log\left(\frac{l}{\beta}\right) + \log\frac{l}{2\pi\varepsilon} + \mathcal{O}(e^{-2\pi l/\beta}) \right), \qquad l \gg \beta, \tag{25}$$

and we see a volume law linear in $l$ appearing as the leading term.

These temperature corrections however no longer have the right scaling behavior when $z \neq 1$, but we use the scale invariant and dimensionless combinations (22) to make an ansatz for the temperature corrections. At small temperatures, we make an ansatz generalizing (24):

$$S = \frac{c}{3}\log\left(\frac{l}{\varepsilon}\right) + f_2(z)\frac{l^2}{\beta^{2/z}} + \mathcal{O}(l^4/\beta^{4/z}), \qquad l \ll \beta^{1/z}, \qquad z \text{ odd}, \tag{26}$$

for some function $f_2(z)$ independent of any scale with $f_2(1) = c\pi^2/18$. This expansion only holds for odd values of $z$, because the leading term (the "area" term at zero temperature) for even values of $z$ is absent. Notice again the absence of a linear term in $l$. This time, there is no a priori reason for it, but our lattice results will establish it. It in fact establishes that, for odd $z$, there are no odd powers of $l\beta^{-1/z}$ for small temperatures.

For even values of $z$, the lattice results show that all powers of $l\beta^{-1/z}$ appear, and we can make a low temperature expansion

$$S = f_1(z)\frac{l}{\beta^{1/z}} + f_2(z)\frac{l^2}{\beta^{2/z}} + \mathcal{O}(l^3/\beta^{3/z}), \qquad l \ll \beta^{1/z}, \qquad z \text{ even}, \tag{27}$$

for some functions $f_{1,2}$. The leading term in this expansion is already a volume law.

Similarly, at large temperatures, we generalize the $z = 1$ result to

$$S = \frac{l}{\beta^{1/z}}\left(g(z) + \frac{S_{off}(z)}{l\beta^{-1/z}} + \mathcal{O}(\beta^{2/z})\right), \qquad l \gg \beta^{1/z}, \tag{28}$$

for some function $g(z)$ with $g(1) = c\pi/3$ and a constant offset correction to the expansion $S_{off}(z)$. It is a non-trivial result that this is the leading term if we don't assume that a volume law should come out at large temperature, as any higher power of $l/\beta^{1/z}$ would be dominant. There can be subleading terms similar as for $z = 1$, such as logarithmic terms, and we include them in the next section. Again, the lattice approach supports the ansatz (28) for both even and odd values of $z$, and in the next section, we give numerical values for $g(z)$ and $S_{off}(z)$.

## 3 Lattice Results

In this section we study the entanglement of Lifshitz fermions on a finite lattice with $N$ lattice sites and lattice spacing $\varepsilon$. We discretize and rescale the localized wave functions $\psi_j \equiv \varepsilon^{1/2}\psi(j\varepsilon)$ to make them dimensionless, and make the plane wave ansatz $\psi_j = c_k(\omega)e^{i(j\varepsilon k+\omega t)}$. Then we discretize the spatial derivative by using the centered difference (to preserve hermiticity of the Lagrangian) limit definition:

$$\partial_1\psi_j = \frac{\psi_{j+1} - \psi_{j-1}}{2\varepsilon} = \frac{1}{2\varepsilon}(e^{ik\varepsilon} - e^{-ik\varepsilon})\psi_j = \frac{i}{\varepsilon}\sin(k\varepsilon)\psi_j. \tag{29}$$

Hence

$$(i\partial_1)^z\psi_j = (-\tilde{k}(k))^z\psi_j, \qquad \text{where} \qquad \tilde{k}(k) = \frac{1}{\varepsilon}\sin(k\varepsilon). \tag{30}$$

Note that this has the right continuum limit $\tilde{k} \to k$ when $\varepsilon \to 0$. The equations of motion in equation (8) yield the dispersion relation

$$\omega_k^2 = \alpha^2\tilde{k}(k)^{2z} + m^2. \tag{31}$$

Notice that when $m = 0$, $\omega_k = \alpha|\tilde{k}|^z$. Furthermore, because of the discretization, the dispersion relation is no longer a monotonic function of $k$, which means that there are in general

two modes associated with a given energy. This phenomenon is know as fermion-doubling and results in a central charge $c = 2$ for the lattice Dirac fermions that is a factor 2 larger than the central charge in the continuum system.

A general solution to the equations of motion is a superposition of plane waves which satisfy boundary conditions with a phase shift: $\psi_N = e^{2\pi\theta i}\psi_0$. This restricts the values of $k$ to

$$k = \frac{2\pi(\theta + \kappa)}{L}, \qquad \text{with } \kappa \in \{0, 1, ..., N-1\}, \qquad \text{and } L = N\varepsilon. \tag{32}$$

As we are interested in the large $N$ limit, whilst keeping $L$ fixed, the value of $\theta$ becomes irrelevant. Without loss of generality we consider periodic boundary conditions. We obtain

$$\psi_{\pm,j} = \frac{1}{\sqrt{2N}} \sum_{\kappa=0}^{N-1} \frac{1}{\sqrt{\omega_k}} e^{ij\kappa\frac{2\pi}{N}} \left( \pm a_k^\dagger e^{i\omega_k t} \sqrt{\omega_k \pm \alpha(-\tilde{k})^z} + b_k e^{-i\omega_k t} \sqrt{\omega_k \mp \alpha(-\tilde{k})^z} \right). \tag{33}$$

We have introduced the annihilation operators $a_k$ and $b_k$, which satisfy the usual equal time anti-commutation relations $\{a_p, a_k^\dagger\} = \delta_{p,k} = \{b_p, b_k^\dagger\}$, which follow from the anti-commutation relations of $\psi$ and the Kronecker delta $\delta_{ij} = \frac{1}{N}\sum_{n=1}^{N} e^{i\frac{2\pi n}{N}n(i-j)}$. This reduces the Hamiltonian to $H = \sum_{\kappa=0}^{N-1} \omega_k(a_k^\dagger a_k + b_k^\dagger b_k - 1)$. For the case where $m = 0$, the dispersion relation is gapless and the term $\omega_k \pm \alpha(-\tilde{k})^z$ vanishes depending on the sign of $\tilde{k}$ and the parity of $z$:

$$\psi_{+,j} = \begin{cases} \frac{1}{\sqrt{2N}} \sum_{\kappa=0}^{N-1} e^{ij\kappa\frac{2\pi}{N}} \left( a_k^\dagger e^{i\omega_k t} \sqrt{1 - \text{sgn}(\tilde{k})} + b_k e^{-i\omega_k t} \sqrt{1 + \text{sgn}(\tilde{k})} \right); & z \text{ odd}, \\ \frac{1}{\sqrt{N}} \sum_{\kappa=0}^{N-1} e^{i(j\kappa\frac{2\pi}{N} + \omega_k t)} a_k^\dagger; & z \text{ even}. \end{cases}$$
$$\psi_{-,j} = \begin{cases} \frac{1}{\sqrt{2N}} \sum_{\kappa=0}^{N-1} e^{ij\kappa\frac{2\pi}{N}} \left( -a_k^\dagger e^{i\omega_k t} \sqrt{1 + \text{sgn}(\tilde{k})} + b_k e^{-i\omega_k t} \sqrt{1 - \text{sgn}(\tilde{k})} \right); & z \text{ odd}, \\ \frac{1}{\sqrt{N}} \sum_{\kappa=0}^{N-1} e^{i(j\kappa\frac{2\pi}{N} - \omega_k t)} b_k; & z \text{ even}. \end{cases} \tag{34}$$

Inverting these relations, we express the $a$ and $b$ operators in terms of the spinor operators. Then for even $z$ one easily verifies that the ground state is equal to the direct product of an occupied $+$-spinor state and empty $-$-spinor over all sites. As a consequence the EE must vanish for even $z$. For $m \neq 0$ this argument no longer holds.

We distill the EE from the two point correlation functions [11, 14]. The EE is given by

$$S = -\sum_{n=1}^{2N} (1 - c_n)\log(1 - c_n) + c_n \log c_n, \tag{35}$$

where $c_n$ is the $n$-th eigenvalue of the correlation matrix restricted to our subsystem, i.e. the matrix constructed by all correlations between the local spinor components. The general equal time two point correlation functions of the spinor components are given by

$$\langle \psi_{\pm,i}^\dagger \psi_{\pm,j} \rangle = \frac{1}{2N} \sum_{\kappa=0}^{N-1} \frac{1}{\omega_k} e^{i\kappa(j-i)\frac{2\pi}{N}} ((1 - \langle N_{a,k} \rangle)(\omega_k \pm \alpha(-\tilde{k})^z) + \langle N_{b,k} \rangle(\omega_k \mp \alpha(-\tilde{k})^z));$$
$$\langle \psi_{+,i}^\dagger \psi_{-,j} \rangle = \langle \psi_{-,i}^\dagger \psi_{+,j} \rangle = \frac{1}{2N} \sum_{\kappa=0}^{N-1} \frac{m}{\omega_k} e^{i\kappa(j-i)\frac{2\pi}{N}} (\langle N_{a,k} \rangle + \langle N_{b,k} \rangle - 1), \tag{36}$$

where we introduced the fermion number operators $N_{a,k} = a_k^\dagger a_k$ and $N_{b,k} = b_k^\dagger b_k$. Note that we are not computing propagators here, i.e. we are not considering a time ordered product. Of particular interest is the massless groundstate of the system, where the above correlators reduce to

$$\langle \psi_{\pm,i}^\dagger \psi_{\pm,j} \rangle = \begin{cases} \frac{1}{2}(1 \pm 1)\delta_{i,j}, & \text{for } z \text{ even}; \\ \frac{1}{2N} \sum_{\kappa=0}^{N-1} e^{i\kappa(j-i)\frac{2\pi}{N}} \left( 1 \mp \text{sign}(\tilde{k}) \right), & \text{for } z \text{ odd}. \end{cases} \tag{37}$$

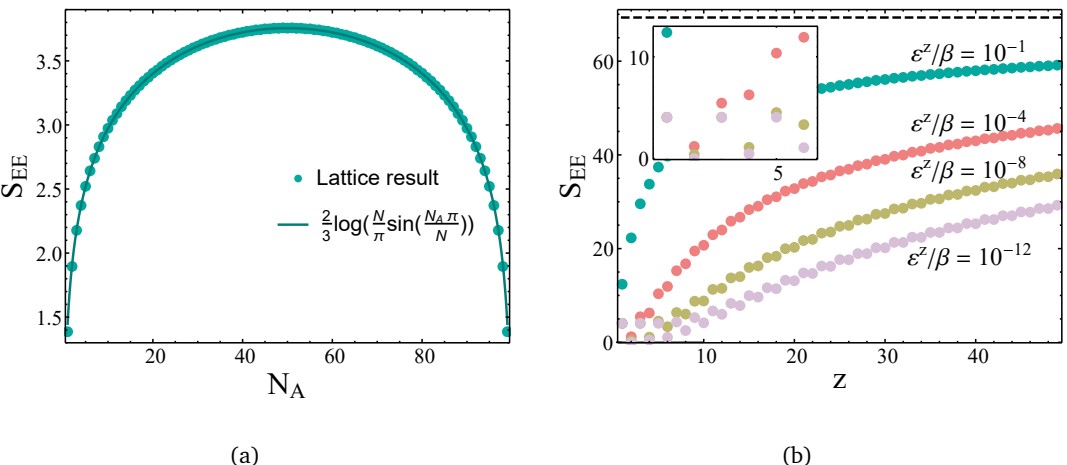

(a)                                   (b)

Figure 2: (Color online) (a) The EE for odd $z$ and zero mass as a function of the segment size $l$ (dots) of the total circular lattice, with $N = 100$ sites, follows the area law (solid line). The value of the central charge is $c = 2$ which is a consequence of the fermion doubling phenomenon. (b) The EE as a function of Lifshitz scaling parameter $z$ for finite temperatures. Note that turning on the temperature brings an explicit $z$ dependence. For large temperatures, as $z$ increases, the EE approaches its maximum value $S_{max} = 2N_A \log 2$ (dashed black line). For small temperature and small $z$ the reminiscences of the zero temperature results are still visible; the parity of $z$ plays an important role (inset). The results are obtained with $N = 1000$ and $N_A = 50$.

Similar to the continuous case, we see that when $m = 0$ all explicit $z$ dependence drops out in these correlation functions, but the correlators still depend heavily on the parity of $z$. In the case that $z$ is even, the EE vanishes. This is due to the fact that the plus spinor correlation sums over all holes but no particles, which yields a Kronecker delta function. The minus spinor correlation sums over all particles, which are not present in the ground state. That is, $c_n = 0$ or $c_n = 1$, which both yield zero EE from equation (35). Note furthermore that we have not yet specified the partitioning of our system. Hence, for even $z$ any partitioning will have vanishing entanglement, whereas for odd $z$, regardless of the partitioning, the entanglement will be independent on the value of $z$. This is a robust consequence of the scaling symmetry of $\psi$, given in equation (4), being independent of $z$.

To connect the result for the $i \neq j$ to the large $N$ limit, we express

$$\frac{1}{2L} \sum_{\kappa=0}^{N-1} e^{2\pi i \frac{x_j - x_i}{L} \kappa} \operatorname{sign}(\tilde{k}) \underset{\substack{N \to \infty \\ \varepsilon \to 0 \\ N\varepsilon = L}}{=} \frac{1}{2L} \sum_{\kappa=0}^{\infty} e^{2\pi i \frac{x_j - x_i}{L} \kappa} = \frac{1}{2} \frac{1}{1 - e^{2\pi i \frac{x_j - x_i}{L}}} \underset{L \to \infty}{=} \frac{-i}{4\pi(x_i - x_j)}, \quad (38)$$

for the continuous system of finite fixed size $L$, followed by the large $L$ limit.

Recall that there is a factor of $\varepsilon$ missing compared to the continuum result, because in this section we made the wavefunction dimensionless. A second expectation to check is the area law result for conformal field theory [13] which is validated in figure 2a. The central charge is 2: Two times the central charge of a continuous Dirac fermion, which is a consequence of the fermion doubling mentioned earlier. The Dirac fermion has a central charge of 1 since it is composed of two Weyl fermions with central charge 1/2.

Instead of considering the groundstate, one could also consider a thermal state. The expectation value of the number operators then is given by the Fermi-Dirac distribution, which

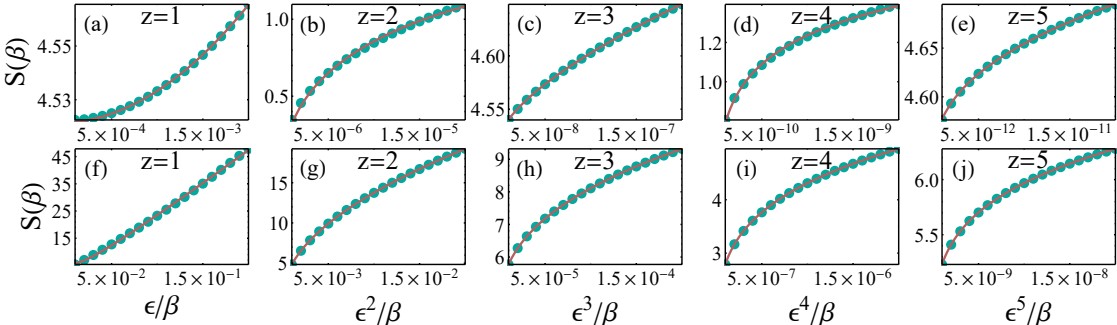

Figure 3: Fits to the functions $S(\beta) = S(\beta = \infty) + f_1(z)l\beta^{-1/z} + f_2(z)l^2\beta^{-2/z}$ in the low temperature regime, and $S(\beta) = S_{off}(z) + g(z)l\beta^{-1/z} + h(z)\log(\varepsilon^z/\beta)$ in the high temperature regime. The results for the fit parameters are in figure 4a. The system size is $N = 10000$ and the subsystem size is $N_A = 100$. For low values of $z$ the high temperature regime is inaccessible due to the finite system size, which bounds the EE as illustrated in figure 4b

reduces equation (36) to

$$\langle\psi_{\pm,i}^\dagger\psi_{\pm,j}\rangle = \frac{1}{2}\delta_{i,j} \pm \frac{1}{2N}\sum_{k=0}^{N-1}e^{ik(j-i)\frac{2\pi}{N}}\frac{\alpha(-\tilde{k})^z}{\omega_k}\tanh\left(\frac{\beta\omega_k}{2\alpha}\right),$$

$$\langle\psi_{\pm,i}^\dagger\psi_{\mp,j}\rangle = -\frac{1}{2N}\sum_{k=0}^{N-1}e^{ik(j-i)\frac{2\pi}{N}}\frac{m}{\omega_k}\tanh\left(\frac{\beta\omega_k}{2\alpha}\right). \tag{39}$$

Note that as $T \to \infty$ the correlation matrix becomes diagonal with maximally degenerate eigenvalue $1/2$. From equation (35) it follows that this maximizes the entropy to its upper bound $2N_A\log 2$, yielding a volume law. In figure 2b the EE is plotted as a function of $z$ for different temperatures and zero mass. For low $z$ the reminiscences of the parity dependence on $z$ (which we explored in the zero temperature regime) are still visible, but they blur out as $z$ increases and the entropy approaches its maximal value. This also follows from equation (39): since $|\tilde{k}| < 1$, we have $\omega_k \to 0$ as $z \to \infty$.

Furthermore, we study the temperature corrections to the area law as a function of $z$ in the high and low temperature regime as suggested in equations (25) to (27) by numerically computing the EE as a function of temperature in both regimes and making fits for each value of $z$. The results are given in figures 3 and 4a. The results again show a strong distinction between even and odd $z$: For even $z$ a linear dependence on $l\beta^{-1/z}$ appears. The high temperature regime is poorly accessible for low $z$ as a consequence of computational power, due to the upper bound of the EE for finite systems (see figure 4b).

# 4 Holographic Entanglement Entropy

In this section we use a method of producing the EE through a combination of tensor networks and holographic methods. First, we introduce briefly the continuous Multi-scale Entanglement Renormalisation Ansatz (cMERA) which produces the elements necessary to calculate the EE via holographic techniques. We note here that this method is only one candidate for producing emergent spaces from field theories, another more recent approach comes from path integral optimization, see e.g. [37, 38]. It would be interesting to test the compatibility of the results that follow with these methods, however that is beyond the scope of this work.

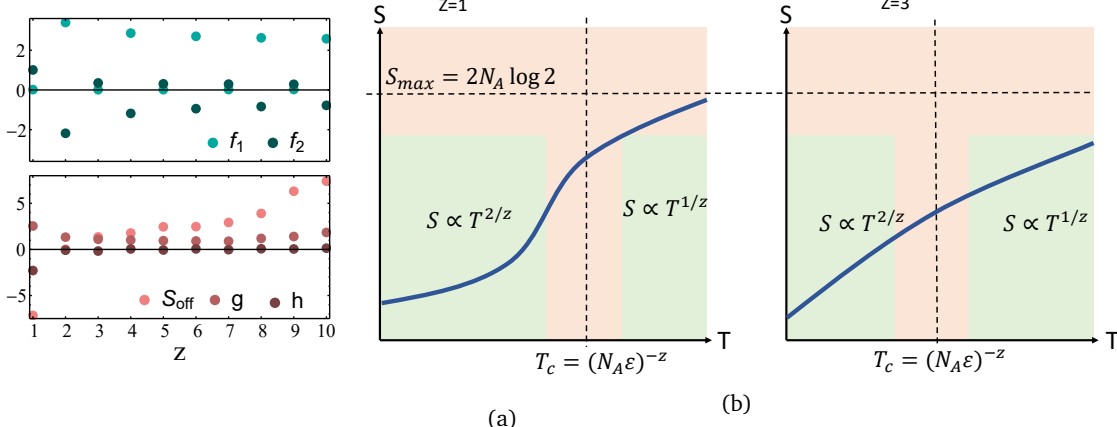

Figure 4: (Color online) (a) Fit parameters as a function of $z$. Note that for even $z$ the leading term in the low temperature regime (blue dots) is linear in $l\beta^{-1/z}$ whereas for odd $z$ it is quadratic, i.e. $f_1 \sim 0$. The high temperature regime (red dots) is poorly accessible for low $z$ as illustrated in (b): For our results there are two relevant bounds when considering the fit to equations (25) to (27): The temperature scale characterizing high and low temperatures $T_c = (\varepsilon N_A)^{-z}$ and the EE saturation limit $S_{max} = 2N_A \log 2$ which is a finite size effect. Since $N_A \leq 100$ is limited by computational capacity, the high temperature regime is poorly accessible for low $z$ (left figure). However, when $z$ increases, $T_c$ decreases such that for high $z$ there is a well accessible regime to fit.

Essentially, one produces a metric element for Anti-de Sitter space from information extracted from the Lifshitz field theory under the cMERA transformation. Using this metric we calculate the area of a minimal surface which in the (1+1)d case is the length of a geodesic on a fixed time slice. The EE of the field theory is then proportional to the size or "area" of this minimal surface by the Ryu-Takayanagi conjecture [39].

## 4.1 Review of (c)MERA

At this point we introduce the ideas involved in bringing MERA into the continuum. This section follows closely the presentation of the introductory work [40] and the subsequent work which is relevant to the calculation of EE in this framework [17]. Before introducing the continuous MERA method it should be made clear which view of the MERA we are taking, which is the perspective of the MERA as a quantum circuit. In this context the MERA is viewed in a "top-down" manner. Starting from an initial unentangled state the state is acted upon by a local unitary operator

$$U_1 = \bigotimes_{j=1}^{N/2} u_{2j-1,2j}, \tag{40}$$

which entangles adjacent sites. In this example local means that the full unitary operator is comprised of 2-site unitary gates or operators. This is followed by a scale transformation so that the lattice spacing and number of spins/qubits/sites are unchanged. We denote this operation by $\mathcal{R}$. It is equivalent to the coarse-graining/isometry step seen in the "bottom-up" picture [41] but modified to be a unitary operation using auxillary qubits. If the depth of the MERA is $\tau = T = \log_2(N)$, as would be the case for a binary MERA scheme, then the output

of the circuit is the state

$$|\Psi_{\text{MERA}}\rangle = U_T \mathcal{R} U_{T-1} \mathcal{R} \ldots \mathcal{R} U_1 |\Omega\rangle \,. \tag{41}$$

The question at this point is how to translate the scale transformation, entangling operation and fiducial state to continuum analogues. In translating to the continuum it is necessary to enforce an ultra-violet cut-off for the field theory, which we denote by: $\Lambda = \varepsilon^{-1}$, where as before $\varepsilon$ is the lattice constant. The Hilbert space defined by the fields with such a cut-off is denoted by $\mathcal{H}_\Lambda$ such that $|\Psi(u)\rangle \in \mathcal{H}_\Lambda$, where $u$ parametrizes the fields and represents the length/energy scale of interest. This parameter is taken such that the momentum $k$ is effectively cut-off as $|k| \leq \Lambda e^u$. In connection to the discrete case, $u$ effectively corresponds to the layer index $\tau$ of the tensor network. By convention we have that $u$ runs over $(-\infty, 0]$, such that the ultraviolet (UV) and infrared (IR) limits are given by: $u_{UV} = 0, u_{IR} \to -\infty$. The states given at these limits are denoted as

$$|\Psi(u_{IR}\rangle \equiv |\Omega\rangle \,, \quad |\Psi(u_{UV})\rangle \equiv |\Psi\rangle \,, \tag{42}$$

such that $|\Omega\rangle$ corresponds to an unentangled reference state and $|\Psi\rangle$ is the ground state in which we compute the EE. Now, as in the lattice implementation we relate a state at any layer or length scale of the MERA to the reference state by a unitary transformation as

$$|\Psi(u)\rangle = U(u, u_{IR}) |\Omega\rangle \,. \tag{43}$$

Likewise an operator, $\mathcal{O}$, can be defined at any scale $u$ as

$$\mathcal{O}(u) \equiv U(0, u)^{-1} \cdot \mathcal{O} \cdot U(0, u), \tag{44}$$

in particular, later, we define the Hamiltonian at different length scales by this action. The form of this unitary operator [17, 40] is

$$U(u_1, u_2) = P \exp\left[ -i \int_{u_2}^{u_1} (\mathcal{K}(u) + \mathcal{L}) \, du \right], \tag{45}$$

where $\mathcal{K}(u)$ and $\mathcal{L}$ are the continuum analogues of the entangling and scaling operations respectively. $P$ denotes a path ordering such that operators are ordered from large to small values of $u$. The scale transformation acting on the IR state leaves it invariant since by definition the IR state is unentangled so each spatial point is uncorrelated with any other point. The entangling operator, $\mathcal{K}(u)$, is designed to generate entanglement but only for modes with wave vectors $|k| \leq \Lambda e^u$. This entanglement generation up to a cut-off is achieved through a function $g(k, u)$ which contains an appropriate cut-off function and the variational parameters, $g(u)$. Generically, $g(k, u)$ is a complex valued function but in this setting it will be real valued. Aside from this, the entangling operator is a quadratic functional of the fields. The following form is taken for the entangling operator [40]

$$\mathcal{K}(u) = i \int dk \left[ g(k, u) \psi_+^\dagger(k) \psi_-(k) + g^*(k, u) \psi_+(k) \psi_-^\dagger(k) \right]. \tag{46}$$

It will be useful in the following discussion to utilise the interaction picture for these unitary operators. This amounts to using

$$U(u_1, u_2) = e^{-iu_1 \mathcal{L}} \cdot P \exp\left( -i \int_{u_2}^{u_1} \tilde{\mathcal{K}}(u) du \right) \cdot e^{iu_2 \mathcal{L}}, \tag{47}$$

where $\tilde{\mathcal{K}}(u) \equiv e^{iu\mathcal{L}} \cdot \mathcal{K}(u) \cdot e^{-iu\mathcal{L}}$. The action of $\mathcal{K}(u)$ is essentially a generalised Bogoliubov transformation of the fields.

While comparisons have been made [42,43] between Anti-de Sitter space and the structure of a MERA network, it has been proposed [17] that by applying a continuous MERA prescription to free field theories one can determine a holographic metric of a space dual to the field theory. In this context, the metric element is given by

$$ds^2 = g_{uu}(u)du^2 + \frac{e^{2u}}{\varepsilon^2}d\vec{x}^2 + g_{tt}(u)dt^2\,. \tag{48}$$

If we consider the ground state of the free field theory then the metric element corresponding to the holographic direction, $g_{uu}$, is related to the variational parameters of the cMERA procedure, $g(u)$, by

$$g_{uu}(u) = \frac{1}{3}g^2(u)\,, \tag{49}$$

for fermionic theories, in the bosonic case $g^2(u)$ appears [17] without a factor of 1/3. The method of determining these parameters is different in both cases. For bosons the variational function $g(u)$ is directly determined from the dispersion relation of the theory. We detail the relation for fermions in the next section. Regardless of this detail, by determining the variational function $g(u)$ using appropriate cMERA methods one may determine a dual metric. Moreover, in the holographic context we compute the EE via the Ryu-Takayanagi proposal [39,44] meaning we do not require information of the time component of the metric here as we calculate on a fixed time slice of the space.

Once we have obtained this metric element we are able to determine the functional form of the EE by calculating the geodesic length for a subsystem $A$ of length $l$ on the boundary provided that one can determine the correct geodesic for the resulting space.

## 4.2 AdS/cMERA Method

Here, we apply the continuous MERA procedure to a free fermionic theory with Lifshitz scaling in $(1+1)$-dimensions. We proceed in a similar fashion to extant literature [17,30,31,40] with the relativistic ($z = 1$) case having appeared in [40]. For this approach we require the Fourier transformed Hamiltonian of the theory. The Hamiltonian here is obtained from the Dirac-Lifshitz Lagrangian equation (3), and has the form:

$$H = -\int dk \left[ \hbar\alpha(-k)^z \left( \psi_+^\dagger\psi_+ - \psi_-^\dagger\psi_- \right) - \mu\alpha^2 \left( \psi_+^\dagger\psi_- + \psi_-^\dagger\psi_+ \right) \right], \tag{50}$$

where the fields are now functions of the momentum. The procedure [40] to find the EE is as follows: firstly an infrared state, $|\Omega\rangle$, is defined by the action of the spinor components on the state. Next the cMERA operator is applied to the Hamiltonian which manifests as a transformation of the fields. Following this, one extremizes the energy functional using the definition of $|\Omega\rangle$ with respect to the variational function $g(u)$ which appears in the definition of the angle that the field transformation depends on. This determines the angle, $\varphi_k$, associated to the true ground state. Having determined $\varphi_k$ we then determine the metric element $g_{uu}(u)$ which depends on the variational function $g(u)$. The final step is to calculate the geodesic length using the metric element found for a particular subsystem.

The reference state $|\Omega\rangle$ is chosen such that

$$\psi_+(x)|\Omega\rangle = 0 = \psi_-^\dagger(x)|\Omega\rangle\,. \tag{51}$$

The cMERA operation on the Hamiltonian amounts to replacing the fields in the Fourier transformed Hamiltonian with the transformed fields such that

$$\tilde{\Psi}(k) = M_k(u)\Psi(e^{-u}k) = e^{-\frac{u}{2}}\begin{pmatrix} \cos(\varphi_k(u)) & -\sin(\varphi_k(u)) \\ \sin(\varphi_k(u)) & \cos(\varphi_k(u)) \end{pmatrix}\Psi(e^{-u}k), \tag{52}$$

where (see App. v1 [40]) the angle is defined as

$$\varphi_k \equiv \lim_{u_{IR} \to -\infty} \int_0^{u_{IR}} du\, g(e^{-u}k, u) = \lim_{u_{IR} \to -\infty} \int_0^{u_{IR}} du\, g(u)\frac{k}{\Lambda}\Gamma\left(\frac{|k|}{\Lambda}\right), \tag{53}$$

where $\Gamma(|k|/\Lambda)$ implements the momentum cut-off and can be taken to be a Heavyside step function, $\Theta(1 - |k|/\Lambda)$. Moreover, by inverting this relation using the Leibniz integral rule we find an expression for $g(u)$ using the form of $g(k, u)$ shown above, the steps involved are presented in [40] which we rederive in appendix A, but the result is that

$$g(u) = \frac{|k|^2}{\Lambda}\partial_{|k|}\left(\frac{\Lambda}{k}\varphi_k\right)\Big|_{|k|=\Lambda e^u} = -\varphi_k + |k|\partial_{|k|}\varphi_k\Big|_{|k|=\Lambda e^u}. \tag{54}$$

After the transformation of the fields, the massive Hamiltonian is given by

$$H = -\int dk\, e^{-u}\{[\hbar\alpha(-k)^z\cos(2\varphi_k(u)) - \mu\alpha^2\sin(2\varphi_k(u))][\psi_+^\dagger(\tilde{k})\psi_+(\tilde{k}) - \psi_-^\dagger(\tilde{k})\psi_-(\tilde{k})]$$

$$- [\hbar\alpha(-k)^z\sin(2\varphi_k(u)) + \mu\alpha^2\cos(2\varphi_k(u))][\psi_+^\dagger(\tilde{k})\psi_-(\tilde{k}) - \psi_-^\dagger(\tilde{k})\psi_+(\tilde{k})]\}, \tag{55}$$

where $\tilde{k} = ke^{-u}$. Now we determine the energy functional, $E[g]$, by evaluating the inner product $\langle\Omega|H|\Omega\rangle$ in the infrared limit. We then obtain the energy functional

$$E[g] = \int dx \int \frac{dk}{2\pi}[\hbar\alpha(-k)^z\cos(2\varphi_k) - \mu\alpha^2\sin(2\varphi_k)]. \tag{56}$$

Subsequently, after taking the functional derivative with respect to the metric function $g(u)$ one finds the condition which minimizes the energy to be

$$\tan(2\varphi_k) = -\frac{m}{(-k)^z}. \tag{57}$$

It should be noted here that this expression is valid for the range of scales $u \in (-\infty, 0]$ and as a result the resulting expression for the angle is valid up to the momentum cut-off, $|k| < \Lambda$. This should not be really thought of as a restriction since the cut-off $\Lambda$ should be taken to infinity in the end. As a result, we have the following expression after use of trigonometric identities:

$$\varphi_k(u) = \frac{1}{2}\arcsin\left[\frac{k^z}{\sqrt{k^{2z} + m^2}}\right] - (-1)^z\frac{\pi}{4}\Big|_{k=\Lambda e^u}. \tag{58}$$

One should keep in mind that here the momentum is set according to $k \to \Lambda e^u$ to obtain the angle and should in this context be seen as a positive quantity. However, as a verification of the lattice result, we look at the massless case here. By taking $m = 0$ at this point the angle becomes a constant, differing only with respect to the parity of $z$ and as such the function $g(u)$ is equal to the angle, $\varphi_k$, up to an overall sign:

$$g(u) = \frac{\pi}{4}((-1)^z - 1). \tag{59}$$

Given the constant $\varphi_k$ value, the entropy calculation becomes rather direct which we produce now. Essentially, for the massless case and $z$-odd, the entropy is found by the calculating the geodesic length using the metric

$$ds^2 = \frac{g^2}{3}du^2 + \frac{e^{2u}}{\varepsilon^2}dx^2\,. \tag{60}$$

Then, using the reparametrization $1/r = e^u/\varepsilon$ and rescaling the $x$ direction by $x \to (\sqrt{3}/g)x$ $\equiv \tilde{x}$, this is a pure AdS metric for $(2+1)$-dimensions on a fixed time-slice

$$ds^2 = \frac{(g^2/3)}{r^2}\left(dr^2 + d\tilde{x}^2\right)\,. \tag{61}$$

The geodesic length is determined using the following parametrization of the curve

$$\gamma = \left\{\tilde{x}(t) = \frac{l}{2}\cos(\pi t), \qquad r(t) = \frac{l}{2}\sin(\pi t)|t \in [0,1]\right\}\,. \tag{62}$$

The length of such a curve is then obtained by calculating

$$|\gamma| = \int_0^1 dt\,\sqrt{g_{\mu\nu}\dot{\gamma}^\mu\dot{\gamma}^\nu} = \frac{2\pi g}{\sqrt{3}}\int_0^{1/2}dt\,\frac{1}{\sin(\pi t)}, \tag{63}$$

$$= -\frac{2g}{\sqrt{3}}\left[-\log(\cos(\pi t/2)) + \log(\sin(\pi t/2))\right]_{t=0}^{t=1/2}\,, \tag{64}$$

$$= -\frac{2g}{\sqrt{3}}\log\left(\sin\left(\frac{\pi\alpha}{2}\right)\right)\,. \tag{65}$$

A cut-off needs to be inserted of $\alpha \ll 1$ on the lower limit of the integration to prevent the integral diverging. The upper limit yields zero, leaving only the result. Considering the parametrization of $r$, we have that for small $t$ near the boundary of the space: $r(t) \sim \frac{l\pi\alpha}{2}$. However, we also have the UV cut-off $r \sim \varepsilon$ meaning that, $\alpha \to 2\varepsilon/\pi l$, and thus we have the result

$$|\gamma| \sim \frac{2g}{\sqrt{3}}\log\left(\frac{l}{\varepsilon}\right)\,. \tag{66}$$

Hence, the EE is given by

$$S_z \propto \frac{2g}{\sqrt{3}}\log\left(\frac{l}{\varepsilon}\right), \tag{67}$$

meaning that:

$$S_z \propto \begin{cases} \frac{\pi}{\sqrt{3}}\log\left(\frac{l}{\varepsilon}\right)\,, & z\text{ odd, where } g = \pi/2\,, \\ 0\,, & z\text{ even, where } g = 0\,. \end{cases} \tag{68}$$

This is in agreement with the results from our correlation function based calculations up to a multiplicative factor. Here that constant would be $c/\pi\sqrt{3}$ which we can determine from comparison to the $z = 1$ known result. Inserting such a factor yields the two cases, distinguished by the parity of $z$:

$$S_z = \begin{cases} \frac{c}{3}\log\left(\frac{l}{\varepsilon}\right)\,, & z\text{ odd}\,, \\ 0\,, & z\text{ even}\,. \end{cases} \tag{69}$$

These two cases are confirmed by the prior results found by calculation using correlation function methods. One should note that the cMERA technology requires additional information to determine the entropy and as yet produces only the functional form of the entropy. In other words, the constant of proportionality in question is not manifestly determined in the cMERA framework.

# 5 Conclusion

We have studied the EE between fermions with a Lifshitz scaling symmetry in both continuous and discrete models. The results are quite different from the results for Lifshitz bosons [29–31]. In the ground state, the most striking difference is that for fermions, there is a strong dependence on the parity of the scaling exponent $z$. For even $z$ and zero mass, the ground-state becomes a pure product state in the spatial spinor representation. Hence, there is no EE. This is reaffirmed by results from the holographic cMERA approach. Other than its parity, the value of $z$ does not affect the EE of the massless ground-state. This indepence on the value of $z$ is a robust consequence of the scaling symmetry of the system and hence extends to any partitioning. Considering the single interval partitioning, we find for odd $z$ that the area law is reproduced (see figure 2a) with a central charge that is twice the value of continuous Dirac fermions due to the fermion doubling on the lattice.

In the thermal state a more explicit dependence on $z$ emerges. However, the parity of $z$ remains a distinguishing factor for low values of $z$ and low temperatures. The low temperature power series expansion of the EE in the scale invariant quantity $l\beta^{-1/z}$ does not contain odd powers for odd $z$, corresponding to the known relativistic result for $z = 1$.

It would be interesting to have better analytic control of the continuum limit, and to extend the analysis to non-integer, continuous values of $z$. Even for the free case that we consider here, we expect this to be a nontrivial extension due to branch cuts in the Lifshitz dispersion relation.

There are various further extensions one can consider, such as the mass deformed case where Lifshitz scale symmetry and chiral symmetry is broken. Also, the presence of interactions and extension to higher dimension are useful. For strongly interacting fermions, one can make contact with Lifshitz holography, for which there are known answers for the EE from the Ryu-Takayanagi formula. We leave this for further study.

# Acknowledgements

K.K. would like to thank Ian Jubb for numerous helpful discussions in the course of completing this work.

**Funding information**    D.H. has received funding from the European Research Council (ERC) under the European Unions Horizon 2020 research and innovation programme (Grant agreement No. 725509). K.K. acknowledges Science Foundation Ireland for financial support through Career Development Award 15/CDA/3240.

# A    Expression for $g(u)$ in terms of $\varphi_k$

Recall the definition of the exact Bogoliubov angle $\varphi_k$ (written as $f(k)$ in the original work [40])

$$\varphi_k \equiv \lim_{u_{IR} \to -\infty} \int_0^{u_{IR}} g(e^{-u}k, u) du. \tag{70}$$

To isolate the relevant function $g(u)$ which comprises the variational part of $g(k, u)$ we need to note the form chosen for $g(k, u)$ in this setting

$$g(k/\Lambda, u) = g(u)\frac{k}{\Lambda}\Gamma\left(\frac{|k|}{\Lambda}\right), \tag{71}$$

where the cut-off function, $\Gamma$, is taken to be the Heavyside step function, $\Theta$[2]. This form of the function is chosen so that the $k$-dependence of $g(k,u)$ is s-wave meaning that it only depends on $|k|$. If instead the (dis)entangler were to depend on the vector $k^i$ then on the holographic side this would correspond to excitations of higher spin fields in the dual higher spin gravity theory. This point is commented on in [17]. Such a dependence on $k$ in $g(k,u)$ would therefore constitute a generalization of current work. One such situation where this may be necessary would be if one were to consider multiple copies of the fermion field thus describing a higher spin theory.

The first step inverting the definition of $\varphi_k$ is to use the change of variables: $z = e^{-u}$ to give

$$
\begin{aligned}
\varphi_k &= \lim_{u_{IR} \to -\infty} \int_1^{e^{-u_{IR}}} g(zk, -\ln(z)) \frac{-1}{z} dz, \\
&= -\int_1^{\infty} g(-\ln(z)) \frac{zk}{\Lambda} \Gamma\left(\frac{z|k|}{\Lambda}\right) \frac{1}{z} dz, \\
&= -\frac{k}{\Lambda} \int_1^{\infty} g(-\ln(z)) \Theta(1 - z|k|/\Lambda) dz, \\
&= -\frac{k}{\Lambda} \int_1^{\frac{\Lambda}{|k|}} g(-\ln(z)) dz, \\
\implies -\frac{\Lambda}{k} \varphi_k &= \int_1^{\frac{\Lambda}{|k|}} g(-\ln(z)) dz.
\end{aligned}
$$

Next, with this relation we differentiate both sides with respect to $|k|$ to remove the integral using the Leibniz integral rule

$$
\frac{d}{dx}\left(\int_{f_1(x)}^{f_2(x)} h(y) dy\right) = h(f_2(x)) f_2'(x) - h(f_1(x)) f_1'(x). \tag{72}
$$

Note that our lower limit is independent of $|k|$ so we have a relatively simple result

$$
\frac{d}{d|k|}\left(\int_1^{\frac{\Lambda}{|k|}} g(-\ln(z)) dz\right) = g\left(-\ln\left(\frac{\Lambda}{|k|}\right)\right) \frac{d}{d|k|}\left(\frac{\Lambda}{|k|}\right). \tag{73}
$$

Combining this rule with our relation to $\varphi_k$ yields

$$
\begin{aligned}
\frac{d}{d|k|}\left(-\frac{\Lambda}{k} \varphi_k\right) &= g\left(-\ln\left(\frac{\Lambda}{|k|}\right)\right)\left(\frac{-\Lambda}{|k|^2}\right), \\
g\left(-\ln\left(\frac{\Lambda}{|k|}\right)\right) &= \frac{|k|^2}{\Lambda} \frac{d}{d|k|}\left(\frac{\Lambda}{k} \varphi_k\right).
\end{aligned}
$$

The final step is to express this relation in terms of $u$ again which amounts to the replacement, $|k| \to \Lambda e^u$

$$
g(u) = \frac{|k|^2}{\Lambda} \frac{d}{d|k|}\left(\frac{\Lambda}{k} \varphi_k\right)\bigg|_{|k|=\Lambda e^u}. \tag{74}
$$

---

[2]This choice of cut-off function is for ease of the calculations, although a smooth function such as $\exp(-|k|^2/\Lambda^2)$ could be chosen to ensure that the entangler $\mathcal{K}(u)$ is local. However, for the purposes of this derivation it is not necessary.

For the version used in the text we expand the RHS

$$
\begin{aligned}
g(u) &= \frac{|k|^2}{\Lambda}\left(\frac{d}{d|k|}\left(\frac{\Lambda}{k}\right)\varphi_k + \frac{\Lambda}{k}\frac{d\varphi_k}{d|k|}\right)_{|k|=\Lambda e^u}, \\
&= \frac{|k|^2}{\Lambda}\left(\left(\frac{-\Lambda}{|k|^2}\right)\mathrm{sgn}(k)\varphi_k + \frac{\Lambda}{k}\frac{d\varphi_k}{d|k|}\right)_{|k|=\Lambda e^u}, \\
&= -\mathrm{sgn}(k)\varphi_k + k\frac{d\varphi_k}{d|k|}\bigg|_{|k|=\Lambda e^u}.
\end{aligned}
$$

For the purpose of determining the metric element, $g_{uu}(u) = g^2(u)/3$, the sign of $k$ is irrelevant as $k$ is set to the positive quantity $\Lambda e^u$ in the end and the total expression of $g(u)$ appears as a squared quantity. As such we write

$$
g(u) = -\varphi_k + |k|\frac{d\varphi_k}{d|k|}\bigg|_{|k|=\Lambda e^u}. \tag{75}
$$

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
