# Peer review of "Entanglement Entropy with Lifshitz Fermions"

_SciPost Physics, doi:SciPost Phys. 11, 031 (2021)_

## Round 2 · Referee Report · Anonymous (Referee 1) · 2021-5-9

Strengths
2 – The main result is derived by two independent methods
Weaknesses
Report
Requested changes
I have noticed the following minor issues (in order of decreasing importance) which I think can be improved, even though they don’t significantly affect the overall scientific quality of the paper:
1 - In the Conclusion section the authors discuss some possible generalisations of their work, (e.g. interactions and higher dimensions), but one issue that I think is interesting is not directly addressed in the paper, and that is the issue of entanglement entropy of unions of multiple intervals.
For even z, if the ground state is a direct product then the result that entanglement entropy vanishes for all even z will of course also extend to unions of multiple disjoint intervals.
For odd z, however, this is less clear to me. In the CFT case (z=1) for example, it is known that equation (19) is generic, while for the union of two or more intervals there is no generic formula valid for all values of c. (Of course, holography provides results in the large c limit)
So, insofar as this doesn’t require too extensive further calculations, I think it would improve the paper to add a discussion of the case for multiple intervals. Do the authors have reason to expect that for odd z, the entanglement entropy of unions of disjoint intervals in the groundstate will also be independent of z? If this is not the case, it might avoid confusion for some readers if this caveat can be added to the abstract or the Introduction section.
2 - In deriving equation (18) from (16), it should be checked whether there is a factor of i missing. At least Mathematica seems to suggests that: $\int_{-\infty }^{\infty } \frac{\text{sgn}(k)+1}{4 \pi }e^{-ikx} \, dk = \frac{\delta (x)}{2}-\frac{i}{2 \pi x}$.
3 - The caption of figure 1 reads in part: “… The lattice system (a and b) has N sites and a lattice spacing ε. The interactions are depicted for z = 1 (a) and z = 2 (b). ”. Given the figure which consists of three parts (a,b,c), I believe “a” should be replaced by “b” and “b” by “c” in these sentences.
4 - In figure 2(a), the entanglement entropy is supposed to be plotted for a lattice with N=1000 points, and this is consistent with the maximum value of the curve ($\gtrsim 3.5$), but the markings 20, 40, 60, 80 on the $N_A$ axis would seem to indicate N=100. So this should be corrected.
5 - On the left-hand side of equation (13), there is one more “(“ than “)” bracket.
6 - On page 3, the authors write: “… present the Lagrangian and for free Lifshitz fermions and we determine the two-point correlator. ” I believe one of the “and” should be deleted.
7 - On page 7: “This expansion only holds for odd values of zero, …”. I assume zero should be replaced with z.
8 - On page 15: “The geodesic is the found from the length of the parametric curve”. Something is off with this sentence, and it should be clarified.
9 - On page 16: “In other words, constant of proportionality in question is not manifestly determined in the cMERA framework.” I assume it should be “the constant”.
10 - It appears to me that the hyperlinks that the pdf of the paper includes to ArXiv versions of papers don’t work. Maybe this is an issue with the bibliography style file that is being used.
Dear referee,
Thank you for your detailed report. In the revised manuscript we have implemented all your requested changes. Moreover, we add more details on the derivation of equation 53 in an appendix.
We especially appreciate the point you raised about unions of multiple intervals. To reply to that: The scaling weight of the fermion is independent of z and so are the correlation functions: they don't depend on z other than via their parity (even or odd), see equation 37. As a consequence we do not expect a z dependence to emerge by any choice of partitioning of the system (unless the Lifhsitz scale symmetry is broken, for instance by finite temperature or mass). This insight will be made more explicit in the revised manuscript.
Kind regards,
Dion Hartmann
On behalf of all authors

Anonymous on 2021-06-24 [id 1521]
In this work authors consider fermionic QFT in 1+1 dimensions with Lifshitz scaling symmetry. First, they analyze ground-state entanglement entropy of a single interval (in the zero mass limit) numerically, focusing on the dependence on the scaling exponent z (that they consider to be an even or odd integer). Interestingly, they find that for even z the state is unentangled and entropy vanishes. For odd z, they analyze finite size and finite temperature entropies and propose certain semi-analytical results in the high- and low-temperature limits.
In the second part they discuss a cMERA approach to holographic geometry where one can extract a hyperbolic geometry from the cMERA circuit. They derive the corresponding metric in which entanglement entropy is again just a geodesic (Like in holography but for free fermion. Simple kinematics of a single interval.) and further support their even as well as odd numerical results using these geometric arguments.
I find this work interesting, new and worth publishing. To improve the paper, I would suggest (and maybe comment):
In this paper: arXiv:1212.1164 [hep-th], authors considered “First Law” for EE (also for non-relativistic theories). Are these new results (say for low temperature) consistent or provide new/different behaviour ?
The cMERA arguments are only valid for free theories and more modern/updated approach to geometry for large-c CFT is based on Path Integrals and Liouville action. See for example: arXiv:2005.11804 [hep-th] for 2+1. It seems that this fermion setup may be good for analysis in 1+1d.
Is it possible to study entanglement for two intervals numerically in this setup?
Finally a small technical detail: What is $N_A$ in (25) computed form (23)? There is no $N_A$ in (23).
Author: Dion Hartmann on 2021-07-02 [id 1541]
(in reply to Anonymous Comment on 2021-06-24 [id 1521])Dear referee,
We thank you for your report and have added clarifiactions to our manuscript based on your suggestions and comments. In this reply we briefly adress them below:
2 & 3. We agree that there are more up-to-date methods than the cMERA method employed here, however, we have not yet explored all of these methods. Nonetheless, if the procedure of the supplied paper holds for general theories and arbitrary $z$ as posited then this would be an interesting avenue for a complementary investigation, especially when combined with numerical approach. We thank you for bringing this work to our attention.
Kind regards, Dion Hartmann On behalf of all authors

---

## Round 2 · Referee Report · Anonymous (Referee 2) · 2021-5-24

Report
The paper is well written and the theoretical discussion is presented in a clear manner. The presented results are original, and the work set the basis for studying EE for fermion fields with Lifshitz scaling symmetry in more general situations. I therefore strongly recommend publication of the manuscript in SciPost Physics. I would like to address the authors some suggestions for changes, and a question about the scope of the manuscript.
Requested changes
1- Figure 1: "the lattice system (a and b) has" $\rightarrow$ "the lattice system (b and c) has". 2- In eq. (25), it is not clear how the constant $N_{A}$ emerges from eq. (23). 3- Below eq. (26): "This expansion only holds for odd values of $\textit{zero}$" $\rightarrow$ "This expansion only holds for odd values of $z$". 4- Above eq. (37): "where the above $\textit{correlaters}$" $\rightarrow$ "where the above correlators". 5- In figure 3, it is plotted $S(T)$ in terms of $\varepsilon^{z}T$. However, the analytic formulas (26-28) are naturally expressed in terms of $l\beta^{-1/z}$. This point worths to be clarified. 6- I would like to ask the authors why the correlator expressions (equations 17-18, 37, 39) are not expected to be invalid for non-integer $z$.
Dear referee,
We first thank you for your report. In light of your comments we have amended some typos and clarified the text in the revised manuscript as a result of points 1 to 5. Regarding the question in point 6 we have added some clarification in the text and a further comment here.
The case of non-integer z brings forth some issues regarding branch cuts, which occur for instance in the dispersion relation (1) when the wavenumber is negative. The choice of how to deal with these cuts may yield different results for the EE. Furthermore, raising an operator to a non-integer power requires some special care as well. This is an interesting topic and we have reserved it for future investigations.
Kind regards,
Dion Hartmann
On behalf of all authors

---

## Round 3 · List of Changes

1. In the introduction between equations 1 and 2 we added a sentence to clarify issues regarding non-integer z.
2. We fixed typos in the caption of figure 1.
3. We fixed a typo on page 3. We fixed a typo in equation 13.
4. We fixed a missing factor -i in equation 18.
5. We fixed a typo in eq 25. We fixed a typo below eq 26. We fixed a typo above eq 37.
6. We added three sentences to the paragraph after equation 37 to remark the universality of a result with respect to the partitioning of our system.
7. We fixed a typo in the caption of figure 2.
8. We fixed a typo in the axis labels of figure 3.
9. We added two sentences to the first paragraph of section 4 to remark other approaches not taken in the present paper.
10. We added equation 46 and two remarks just before it to clarify on the derivation of g.
11. We added some clarifying remarks below equation 53 and refer to the appendix to be more self-contained with regards to the derivation of g.
12. We added a clarification regarding the geodesic above equation 62.
13. We fixed a typo in the last sentence of section 4.
14. We added a remark to the first paragraph of section 5 highlighting again the universality with respect to the partitioning.
15. We added an appendix to be more self-contained with regards to the derivation of g.
16. We abbreviated several occurences of entanglement entropy to EE.

---

## Editorial Decision

published